# Noise-Robust Fine-Tuning of Pretrained Language Models via External Guidance

**Song Wang**
University of Virginia
sw3wv@virginia.edu

**Zhen Tan**
Arizona State University
ztan36@asu.edu

**Ruocheng Guo**
ByteDance Research
ruocheng.guo@bytedance.com

**Jundong Li**
University of Virginia
jundong@virginia.edu

## Abstract

Adopting a two-stage paradigm of pretraining followed by fine-tuning, Pretrained Language Models (PLMs) have achieved substantial advancements in the field of natural language processing. However, in real-world scenarios, data labels are often noisy due to the complex annotation process, making it essential to develop strategies for fine-tuning PLMs with such noisy labels. To this end, we introduce an innovative approach for fine-tuning PLMs using noisy labels, which incorporates the guidance of Large Language Models (LLMs) like ChatGPT. This guidance assists in accurately distinguishing between clean and noisy samples and provides supplementary information beyond the noisy labels, thereby boosting the learning process during fine-tuning PLMs. Extensive experiments on synthetic and real-world noisy datasets further demonstrate the superiority of our framework over the state-of-the-art baselines.

## 1 Introduction

In recent years, the development of language models has significantly expanded the applications within the field of natural language processing (NLP). Fine-tuning Pretrained Language Models (PLMs) like BERT (Devlin et al., 2018) for specific downstream tasks has become an essential step in real-world implementations (Alt et al., 2020; Wang et al., 2021). In general, achieving significant performance gains in fine-tuning PLMs necessitates the availability of task-specific data for fine-tuning (Zhou and Chen, 2021; Wang et al., 2022). However, obtaining high-quality labeled datasets for this purpose poses significant challenges due to the expensive, complex, and labor-intensive nature of the annotation process (Yu et al., 2019; Bae et al., 2022). For example, large-scale datasets, often derived from web-crawling (Li et al., 2017; Song et al., 2019) or crowd-sourcing (Yan et al., 2014; Williams et al., 2017; Sakaguchi et al., 2021), frequently suffer from the presence of noisy labels.

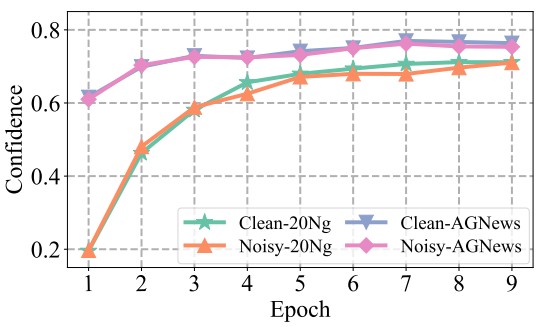

Figure 1: The average confidence of clean and noisy samples during fine-tuning BERT on datasets 20Ng and AGNews with 20% noisy labels.

Prior research (Arpit et al., 2017; Cheng et al., 2021; Zhang et al., 2021c; Wang et al., 2023a) has shown that PLMs are prone to overfitting and generally deliver subpar performance when fine-tuned on datasets containing label noise. Since fine-tuning involves incorporating supervision information to enhance performance on downstream tasks, the presence of noisy labels will mislead the training process and significantly impede the efficacy of PLMs (Zhu et al., 2022). Therefore, there is an immediate need to design effective algorithms for fine-tuning PLMs in the presence of noisy labels.

In the context of learning from noisy labels, an intuitive approach is to separate clean samples from the noisy ones in the training set for model training (Han et al., 2018). For the remaining noisy samples, a prevalent strategy is to pseudo-label them based on model predictions to reduce the adverse impact of noise (Berthelot et al., 2019; Sohn et al., 2020). However, it remains challenging when applying this paradigm to PLMs. This is because PLMs, equipped with prior knowledge encoded in a large size of parameters, tend to easily memorize noisy samples early on during the fine-tuning process. As illustrated in Figure 1, PLMs exhibit similar prediction confidences for both clean and noisy samples, which will result in two challenges

when utilizing existing methods: (1) Existing approaches often rely on confidences generated by models trained on potentially noisy samples (Li et al., 2020; Karim et al., 2022). However, as shown in Figure 1, it is challenging to accurately distinguish clean and noisy samples solely based on confidences. (2) Existing methods are susceptible to erroneous information during pseudo-labeling, as they directly utilize model predictions as pseudo-labels, which can be detrimental when the predictions are inaccurate. PLMs, being capable of easily remembering noisy samples, further exacerbate the risk of capturing and amplifying erroneous information during pseudo-labeling. For example, in the 20Ng dataset (Lang, 1995), when an "autos" sample is assigned a wrong label "hardware", PLMs can easily memorize the erroneous information and thus cannot infer the correct labels in predictions, resulting in incorrect pseudo-labeling.

To overcome these challenges, we propose a novel framework, named **LAFT**, that harnesses the power of **L**arge L**A**nguage Models (LLMs) for **F**ine-**T**uning PLMs. We leverage LLMs trained on extensive corpora and fine-tuned with human instructions, such as GPT4 (OpenAI, 2023), to obtain external guidance in the form of confidence values. Our approach addresses the first challenge of separating reliable samples by utilizing LLM-generated confidences for each class of all training samples. By comparing these confidences with the assigned labels (i.e., the given noisy labels) of training samples, we categorize them into three disjoint subsets: the Easy Clean (EC) Set, the Hard Clean (HC) Set, and the True Noisy (TN) Set. Regarding the second challenge, we propose a novel method to incorporate LLM-generated confidence scores as robust supervision information for all samples to ensure that PLMs learn useful information from them. As LLM-generated confidences are not affected by label noise, they can provide potentially relevant labels that are useful even if they are not entirely accurate. In summary, our contributions are as follows:

- We are the first to explore the potential of leveraging supervision information (i.e., confidence scores) generated by LLMs to tackle the noisy label problem in fine-tuning PLMs.

- We propose a novel framework LAFT that can effectively separate clean and noisy samples and learn from noisy labels, based on the LLMs-generated confidences.

- We conduct extensive experiments on synthetic and real-world noisy datasets and demonstrate the superiority of our framework in fine-tuning PLMs with noisy labels.

## 2 Related Work

### 2.1 Learning from Noisy Labels

Various strategies for learning from noisy labels have been proposed, falling primarily into three categories. The first category is *Sample Selection* methods, which typically employ loss or confidences to identify reliable samples for model optimization. These methods generally necessitate a predefined threshold (Li et al., 2021; Karim et al., 2022) or prior knowledge concerning the noise label rate (Han et al., 2018; Yu et al., 2019) to choose the instances. The second category, dubbed *Label Transition* methods, aims to learn (Tanaka et al., 2018) or generate pseudo-labels (Zhang et al., 2021c; Sohn et al., 2020) to replace the original noisy labels. Lastly, *Regularization* methods design robust loss functions (Liu et al., 2020; Englesson and Azizpour, 2021) or regularization techniques (Zhang and Sabuncu, 2018) that can effectively utilize all samples to enhance the model robustness against label noise. Nevertheless, these methods generally do not consider the scenario of fine-tuning a pretrained model with noisy labels.

### 2.2 Confidence-Guided Sample Separation

To separate noisy samples, existing works leverage the loss or confidences that provide insights into the model's prediction behavior as training proceeds (Yu et al., 2019; Karim et al., 2022). In the context of learning from noisy labels, the key concept is to leverage these dynamics as criteria for identifying and separating noisy samples. Several works propose to identify samples with lower training loss as the clean subset (Li et al., 2020; Han et al., 2018; Jiang et al., 2018; Zhao et al., 2022; Wang et al., 2023b), however, they are generally simplistic and inflexible, resulting in the selection of only easy samples. To address this limitation, alternative approaches have been proposed to effectively utilize the loss or confidences during training, as demonstrated in (Zhang et al., 2021a) and (Nishi et al., 2021). In contrast to existing methods that only rely on confidences generated by PLMs, we leverage LLMs as external guidance that provides more precise separations for fine-tuning PLMs.

## 3 Problem Definition

In this work, we study the problem of fine-tuning PLMs for text classification with noisy labels. Formally, consider a noisy training dataset containing $n$ samples: $\mathcal{D}_{tr} = \{(x_i, \widetilde{y}_i), i = 1, 2, \ldots, n\}$, where $x_i$ is an input sample and $\widetilde{y}_i$ denotes the *assigned* label of $x_i$. Note that $\widetilde{y}_i$ is potentially corrupted, and we denote the true label of $x_i$ as $\overline{y}_i$, which is inaccessible during fine-tuning. Specifically, we aim to fine-tune PLMs with samples in $\mathcal{D}_{tr}$ to achieve satisfactory prediction performance on test samples, while utilizing LLMs as external guidance. Notably, the LLMs remain fixed in our framework, which means we can also use black-box LLMs. In practice, we implement PLMs for text classification by attaching an additional classifier, which will take the output of the PLM as input and generate class probabilities for classification.

## 4 Methodology

The overall framework of LAFT is illustrated in Fig. 2. In particular, we propose to divide all training samples into three subsets based on the accordance among LLM-generated confidences, PLM-generated confidences, and the assigned labels of samples. In particular, we query an LLM to provide confidences for each training sample, spanning all classes. Combining confidences obtained from both LLMs and PLMs, we perform two steps of separation to segregate the entire training set into three subsets: Easy Clean (EC) Set, Hard Clean (HC) Set, and True Noisy (TN) Set. Each of these subsets, displaying unique behaviors, is then subjected to specific noise-robust fine-tuning strategies.

### 4.1 Confidences

Existing studies establish a correlation between confidences and the degree to which deep models memorize specific samples during training (Arpit et al., 2017; Karim et al., 2022). It has been observed that as the model's memorization of a particular sample strengthens, the model tends to assign higher confidence for this sample (Li et al., 2023). Therefore, these methods generally employ confidences to distinguish between clean and noisy samples based on the assumption that the model cannot easily memorize noisy samples (Pleiss et al., 2020; Swayamdipta et al., 2020). However, applying these strategies to fine-tuning PLMs is suboptimal, as PLMs also present high confidence for noisy samples even in the early stage of fine-tuning, as

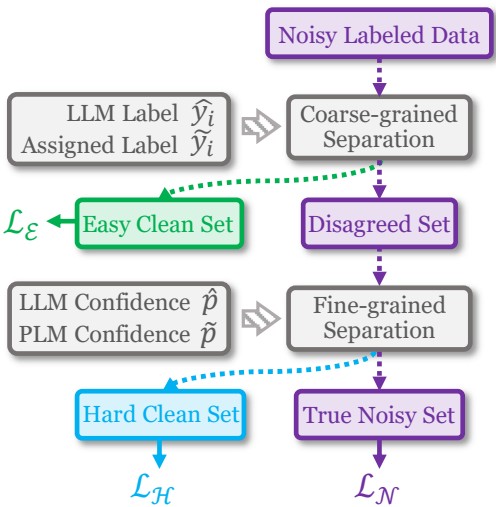

Figure 2: The detailed process of our framework LAFT. We perform two steps of separation to divide all training samples into three subsets with different degrees of label noise: Easy Clean Set $\mathcal{E}$, Hard Clean Set $\mathcal{H}$, and True Noisy Set $\mathcal{N}$. We further propose three different losses to effectively learn from them: $\mathcal{L}_\mathcal{E}$, $\mathcal{L}_\mathcal{H}$, and $\mathcal{L}_\mathcal{N}$.

shown in Fig. 1. To deal with this issue, we propose the utilization of external guidance, in the form of confidences generated by LLMs. Before delving into the specifics of our framework, we provide a formal definition of confidences. Denote the output of the final layer (i.e., the classifier) in PLMs for sample $x_i$ as $\mathbf{z}(x_i) \in \mathbb{R}^N$, where $N$ is the number of classes. The confidence of $x_i$ for the $j$-th class $c_j$ can be represented as follows:

$$\widetilde{p}(c_j; x_i) = \frac{\exp\left(z(c_j; x_i)\right)}{\sum_{k=1}^{N} \exp\left(z(c_k; x_i)\right)}, \quad (1)$$

where $z(c_j; x_i) \in \mathbb{R}$ is the $j$-th value in $\mathbf{z}(x_i)$. Notably, the confidences are obtained from $\mathbf{z}(x_i)$ after a softmax function and thus sum up to one.

Although LLM-generated confidences can provide external guidance, they are not completely accurate. Thus, We conduct a two-step sample separation process based on both LLM-generated and PLM-generated confidences, with the second step providing a more granular distinction.

### 4.2 Coarse-Grained Separation

For the first step of separation, we aim to select samples that are easy to be identified as clean data with guidance from LLMs. Thus, we perform *Coarse-grained Separation*, utilizing confidences generated by LLMs with the raw text data included in the prompt. Here we provide an example of the prompt for querying the LLM to obtain the confidence value for each class:

> Classify the following content: $\{input\ text\}$.
> Select the label from $\{Class\ 1\}$, $\{Class\ 2\}$,
> $\ldots$, $\{Class\ N\}$ and output a confidence value
> for each of them.

We denote the LLM-generated confidence for sample $x_i$ regarding class $c_j$ as $\widehat{p}(c_j; x_i)$, where $c_j \in \mathcal{C} = \{c_1, c_2, \ldots, c_N\}$, and $\mathcal{C}$ is the class set. Then the label obtained by LLM can be represented as

$$\widehat{y}_i = \underset{j=1,2,\ldots,N}{\operatorname{argmax}} \widehat{p}(c_j; x_i). \tag{2}$$

To perform coarse-grained separation, we first provide an assumption as the justification:

**Assumption 1.** *Samples whose assigned labels are the same as LLM-generated labels (i.e., $\widehat{y}_i = \widetilde{y}_i$) can be considered almost clean.*

Note that Assumption 1 is empirically verified in Table 5 in Sec. 5.5. The underlying intuition is that the probability of an LLM-generated label being identical to the assigned label and inconsistent with the ground-truth label is significantly low and thus negligible. In concrete, this assumption allows us to segregate the training samples into two distinct subsets based on the concordance between the LLM-generated label $\widehat{y}_i$ and the assigned (potentially noisy) label $\widetilde{y}_i$ of $x_i$. More specifically, we define the resulting two subsets, the Easy Clean (EC) Set $\mathcal{E}$ and the Disagreed Set $\mathcal{D}$, as follows:

$$\mathcal{E} = \{x_i, \widetilde{y}_i | \widehat{y}_i = \widetilde{y}_i\}, \ \mathcal{D} = \{x_i, \widetilde{y}_i | \widehat{y}_i \neq \widetilde{y}_i\}. \tag{3}$$

It is naturally satisfied that $\mathcal{E} \cup \mathcal{D} = \mathcal{D}_{tr}$ and $\mathcal{E} \cap \mathcal{D} = \emptyset$, where $\mathcal{D}_{tr}$ is the training set. Since the samples in $\mathcal{E}$ are already clean, we can directly fine-tune PLMs using LLM-generated labels $\widehat{y}_i$ based on the cross-entropy loss as follows:

$$\mathcal{L}_{\mathcal{E}} = -\frac{1}{|\mathcal{E}|} \sum_{i=1}^{|\mathcal{E}|} \sum_{j=1}^{N} \widehat{y}_{i,j} \log \widetilde{p}(c_j; x_i), \tag{4}$$

where $\widetilde{p}(c_j; x_i)$ is the PLM-generated confidence for $x_i$ regarding the $j$-th class $c_j$. Here $\widehat{y}_{i,j} = 1$ if $\widehat{y}_i = c_j$, and $\widehat{y}_{i,j} = 0$, otherwise.

### 4.3 Fine-Grained Separation

It is noteworthy that, however, the samples in $\mathcal{D}$ are not completely noisy. This is because the LLM-generated labels are not perfectly correct, as shown in Table 5. Thus, the samples that are clean but incorrectly classified by LLMs will still be categorized into $\mathcal{D}$. Therefore, although we can learn from samples in $\mathcal{E}$ directly with their LLM-generated label $\widetilde{y}_i$, it is still challenging to learn from samples in $\mathcal{D}$, which are only partially clean. Therefore, we further propose to separate the samples in the disagreed set $\mathcal{D}$ into two subsets: the Hard Clean (HC) Set $\mathcal{H}$ and the True Noisy (TN) Set $\mathcal{N}$, referred to as fine-grained separation.

The intuition is that within the disagreed set $\mathcal{D}$, the LLM-generated labels can be incorrect for specific hard samples with correct assigned labels, referred to as Hard Clean (HC) samples. Specifically, the *ideal* separation for $\mathcal{D}$ is as follows:

$$\begin{aligned} \mathcal{H}^* &= \mathcal{D} \cap \{x_i, \widetilde{y}_i | \widetilde{y}_i = \overline{y}_i\}, \\ \mathcal{N}^* &= \mathcal{D} \cap \{x_i, \widetilde{y}_i | \widetilde{y}_i \neq \overline{y}_i\}, \end{aligned} \tag{5}$$

where $\overline{y}_i$ is the true label of $x_i$. Note that although this ideal separation can be completely precise for separating noisy samples in $\mathcal{D}$, it is infeasible in practice, as the true labels are unknown. Therefore, to precisely separate the true noisy samples in $\mathcal{D}$, we propose two thresholds for LLM-generated and PLM-generated confidences, respectively.

In order to achieve more robust LLM-generated confidences distributed over the label space $\mathcal{C}$, we adopt $M$ different augmentations for each input sample to encourage input diversity while keeping the semantics immutable. Denote the augmented samples of $x_i$ as $v_m(x_i)$, where $m = 1, 2, \ldots, M$, and $M$ is the number of augmentations. We can obtain the $M$ LLM-generated confidences as $\widehat{p}(c_j; v_m(x_i))$ for $x_i$ regarding the $j$-th class $c_j$, where $m = 1, 2, \ldots, M$ and $j = 1, 2, \ldots, N$. We aggregate the LLM-generated confidences via the $M$ augmentations as follows:

$$\widehat{p}_a(c_j; x_i) = \frac{1}{M} \sum_{m=1}^{M} \widehat{p}(c_j; v_m(x_i)), \tag{6}$$

where $v_m(x_i)$ is the input example after applying the $m$-th augmentation. As LLM-generated confidences remain fixed during fine-tuning PLMs, we adopt a fixed threshold $\widehat{\tau}$ for fine-grained separation based on $\widehat{p}_a(c_j; x_i)$.

On the other hand, however, PLMs-generated confidences for each sample will change as the fine-tuning proceeds, which results in subpar separation performance if we adopt a fixed threshold to select high-confidence samples as clean ones. Intuitively, the confidences should be lower for HC samples at the beginning of fine-tuning, as the model cannot

easily fit these hard samples within several epochs. Nonetheless, the noisy samples are relatively easier to achieve higher confidence, and thus their confidences will easily achieve higher at the beginning. Therefore, we propose an adaptive threshold $\widetilde{\tau}(t)$ that will increase as fine-tuning proceeds while not reaching an excessively high value:

$$\widetilde{\tau}(t) = \widetilde{\tau} - \exp(-\lambda t), \qquad (7)$$

where $\lambda$ and $\widetilde{\tau}$ are hyper-parameters that control the value of threshold $\widetilde{\tau}(t)$ as fine-tuning proceeds. $t$ denotes the current number of fine-tuning epochs.

Combining the two thresholds, we can perform fine-grained separation by selecting the HC set $\mathcal{H}$ as follows:

$$\begin{aligned} \mathcal{H} = \mathcal{D} \cap \{x_i, \widetilde{y}_i | \max_{c \in \mathcal{C}} \widehat{p}_a(c; x_i) < \widehat{\tau}\} \\ \cap \{x_i, \widetilde{y}_i | \max_{c \in \mathcal{C}} \widetilde{p}(c; x_i) < \widetilde{\tau}(t)\}, \end{aligned} \qquad (8)$$

where $\widehat{\tau}$ and $\widetilde{\tau}(t)$ are the thresholds for LLM-generated confidences ($\widehat{p}_a(c_j; x_i)$) and PLMs-generated confidences ($\widetilde{p}(c_j; x_i)$), respectively. In this manner, the process of separating HC and TN samples, i.e., fine-grained separation, can benefit from both the LLMs and PLMs. Then the remaining samples are categorized into the True Noisy (TN) set $\mathcal{N}$:

$$\mathcal{N} = \mathcal{D} \setminus \mathcal{H}. \qquad (9)$$

### 4.4 Learning from the Hard Clean (HC) Set

Now we have divided the disagreed set $\mathcal{D}$ into $\mathcal{H}$ and $\mathcal{N}$. Recall that ideally, samples in $\mathcal{H}$ are hard yet correct. Thus, for these samples, we can directly utilize their assigned labels $\widetilde{y}$ as training labels. Nevertheless, since the fine-grained separation of $\mathcal{H}$ and $\mathcal{N}$ cannot be perfect, the samples in $\mathcal{H}$ may still be noisy. As both LLMs and PLMs fail to provide a confident prediction for samples in $\mathcal{H}$, we propose to prioritize the assigned label while also incorporating additional information from LLMs and PLMs. Specifically, we employ a weighted loss based on cross-entropy. The weight is enlarged if the summed confidence of the LLM and the PLM is high. We define the loss as follows:

$$\mathcal{L}_{\mathcal{H}} = -\frac{1}{|\mathcal{H}|} \sum_{i=1}^{|\mathcal{H}|} \sum_{j=1}^{N} (\widetilde{y}_{i,j} + \phi_{i,j}) \log \widetilde{p}(c_j; x_i), \qquad (10)$$

where $\widetilde{p}(c_j; x_i)$ is the PLM-generated confidence for $x_i$ regarding the $j$-th class $c_j$. Here $\widetilde{y}_{i,j} = 1$ if $\widetilde{y}_i = c_j$, and $\widetilde{y}_{i,j} = 0$, otherwise. $\phi_{i,j}$ acts as a

weight adjustment that increases when the sum of LLM confidence $\widehat{p}_a(c_j; x_i)$ and PLM confidence $\widetilde{p}(c_j; x_i)$ is larger. Intuitively, if the LLM and PLM are with high confidence regarding a specific class, then the information in their confidence can be useful as they are more likely to be correct. Therefore, we define $\phi_{i,j}$ as follows:

$$\phi_{i,j} = \max\left(\widehat{p}_a(c_j; x_i) + \widetilde{p}(c_j; x_i) - \alpha\widetilde{\tau}(t), 0\right), \qquad (11)$$

where $\alpha > 1$ is a hyper-parameter that controls the threshold $\alpha\widetilde{\tau}(t)$ for $\phi_{i,j}$. As such information can still be inaccurate, we subtract it by the threshold $\alpha\widetilde{\tau}(t)$ to control its magnitude, such that $\phi_{i,j}$ also acts as an adaptive loss weight for these samples.

### 4.5 Learning from the True Noisy (TN) Set

After the fine-grained separation, most samples in the True Noisy (TN) Set should be identified as noisy ones. Nevertheless, it is still challenging for PLMs to use their output as pseudo-labels for fine-tuning, as the prediction errors will accumulate and affect subsequent pseudo-labeling results. Fortunately, the confidences generated by LLMs can provide additional guidance to identify the potential labels for samples in the TN set. We first provide Remark 1 to justify the effectiveness of using LLM-generated labels for optimization.

**Remark 1.** *LLM-generated labels on the True Noisy Set preserve the same accuracy as that on the whole dataset, as LLMs are not affected by noise.*

Remark 1, as empirically verified in Sec. 5.5, demonstrates that even when we categorize most noisy samples into True Noisy Set, the LLM-generated labels can still provide decent guidance without sacrificing accuracy. Given the confidences provided by LLMs, we employ a loss that can enable the PLMs to benefit from them. Specifically,

$$\begin{aligned} \mathcal{L}_{\mathcal{N}} = &-\frac{1}{|\mathcal{N}|} \sum_{i=1}^{|\mathcal{N}|} \sum_{j=1}^{N} \widehat{p}_a(c_j; x_i) \log \widetilde{p}(c_j; x_i) \\ &-\frac{1}{|\mathcal{N}|} \sum_{i=1}^{|\mathcal{N}|} \delta(x_i) \cdot \max_{c \in \mathcal{C}} \widetilde{p}(c; x_i) \log \max_{c \in \mathcal{C}} \widetilde{p}(c; x_i). \end{aligned} \qquad (12)$$

Here in the first term, we leverage the confidences generated by LLMs to learn from potentially correct labels. This is because although LLMs cannot completely predict the correct labels, the confidences still preserve the potentially useful information in other incorrect but relevant labels. Such

benefits cannot be provided by pseudo-labeling, which tends to output a definitive label. For the second term, we utilize the model predictions to exploit useful information from PLMs. The intuition is that, with the relevant label information provided by LLMs, the PLMs can learn accurate label information from the noisy samples in TN set. Consequently, if the model output tends to be confident on specific samples, we can utilize the prediction to further enhance the learning from it. Thus, we further set a threshold for this term, i.e., $\delta(x_i)$, defined as follows:

$$\delta(x_i) = \begin{cases} 1, & \text{if } \max_{c \in \mathcal{C}} \widetilde{p}(c; x_i) > \beta \widetilde{\tau}(t), \\ 0, & \text{otherwise,} \end{cases} \quad (13)$$

where $\widetilde{\tau}(t)$ is computed by Eq. (7). To reduce the effect of confirmation bias, we multiply $\widetilde{\tau}(t)$ by $\beta$, a hyper-parameter that controls the final adaptive threshold, i.e., $\beta\widetilde{\tau}(t)$.

### 4.6 Fine-tuning Objective

After we separate all training samples into $\mathcal{E}$, $\mathcal{H}$, and $\mathcal{N}$, we can combine the three individual losses for them for PLM fine-tuning. Our final fine-tuning objective can be represented as follows:

$$\mathcal{L} = \mathcal{L}_\mathcal{E} + \lambda_\mathcal{H}\mathcal{L}_\mathcal{H} + \lambda_\mathcal{N}\mathcal{L}_\mathcal{N}, \quad (14)$$

where $\lambda_\mathcal{H}$ and $\lambda_\mathcal{N}$ are hyper-parameters that control the importance of $\mathcal{L}_\mathcal{H}$ and $\mathcal{L}_\mathcal{N}$, respectively.

## 5 Experiments

### 5.1 Experimental Settings

**Datasets**. To evaluate the performance of our framework, we first conduct experiments on two synthetic-noise datasets: 20Ng (Lang, 1995) and AGNews (Li and Roth, 2002; Zhang et al., 2015). Following existing works on learning from noisy labels (Patrini et al., 2017; Yao et al., 2020; Zhuang et al., 2023), we adopt three types of synthetic label noise: (1) **Symmetric Noise (SN)** uniformly changes labels to other classes (Han et al., 2018; Xia et al., 2021). (2) **Asymmetric Noise (ASN)** changes labels to other similar classes (Tanaka et al., 2018; Bae et al., 2022). (3) **Instance-Dependent Noise (IDN)** changes labels based on a probability in proportion to the specific sample features (Cheng et al., 2021; Yao et al., 2021). Moreover, we further conduct experiments on three real-world datasets with noisy labels: SemEval (Zhou et al., 2020), TREC (Awasthi et al., 2020), and

Hausa (Hedderich et al., 2020). More details about these datasets are provided in Appendix B.

**Baselines**. We compare our framework to state-of-the-art baselines for learning from noisy labels. In particular, we compare to (1) **Base** (Devlin et al., 2018) that performs fine-tuning with standard cross-entropy loss; (2) **Regularization Methods**: Mixup (Zhang et al., 2018) and GCE (Zhang and Sabuncu, 2018); (3) **Sample-selection Methods**: Co-teaching (Han et al., 2018), Co-teaching+ (Yu et al., 2019), JoCoR (Wei et al., 2020), CR (Zhou and Chen, 2021), and NPC (Bae et al., 2022). Additional details are provided in Appendix C.

**Implementation Details**. We use BERT (Devlin et al., 2018) as the text encoder for all datasets except Hausa, for which we use mBERT. The classifier is implemented as a fully-connected layer and randomly initialized at the beginning, while both the encoder and the classifier will be updated via gradient descent during fine-tuning. All experiments are evaluated on a clean test set, and the average accuracy along with the standard deviation over ten runs is reported for each dataset. We provide more details about the implementation in Appendix D, and our code is provided at https://github.com/SongW-SW/LAFT.

### 5.2 Comparison on Synthetic Datasets

In this subsection, we compare our framework with other baselines on synthetic datasets 20Ng and AGNews, considering different noise types and ratios. Specifically, for Symmetric Noise (SN), we conduct experiments on three different noise rates: 20%, 40%, and 60%. For the other two types of noise, i.e., Asymmetric Noise (AN) and Instance-Dependent Noise (IDN), we adopt two noise rates: 20% and 40%. We present the results in Table 1, 2, and 3. The key observations drawn from the outcomes are as follows: (1) Regardless of the noise type, our LAFT framework persistently surpasses other state-of-the-art baselines, thus showcasing its effectiveness in fine-tuning PLMs with noisy labels. (2) LAFT's performance improvement over other baselines is slightly more pronounced on the 20Ng dataset compared to AGNews. This is attributable to 20Ng containing a greater number of classes ($N = 20$) as opposed to AGNews ($N = 4$). Consequently, our strategy of utilizing LLM-generated confidences can capitalize on the similar labels within 20Ng for PLM fine-tuning, even if LLM predictions are not entirely accurate.

Table 1: The overall performance of various models on synthetic noisy datasets 20Ng and AGNews with Symmetric Noise (SN), where accuracy and standard deviation are reported in %, and the best results are in **bold**.

| Dataset | 20Ng | | | AGNews | | |
|---|---|---|---|---|---|---|
| SN Ratio | 20% | 40% | 60% | 20% | 40% | 60% |
| Base | $79.15 \pm 0.29$ | $71.74 \pm 0.07$ | $61.87 \pm 1.58$ | $81.77 \pm 0.33$ | $78.33 \pm 0.47$ | $73.07 \pm 1.49$ |
| Mixup | $79.10 \pm 0.21$ | $70.58 \pm 0.22$ | $62.48 \pm 2.04$ | $81.25 \pm 0.02$ | $78.24 \pm 0.77$ | $73.12 \pm 1.58$ |
| GCE | $79.56 \pm 0.39$ | $70.46 \pm 0.42$ | $62.61 \pm 1.01$ | $82.98 \pm 0.20$ | $78.46 \pm 0.20$ | $72.75 \pm 0.27$ |
| Co-teaching | $76.68 \pm 0.86$ | $69.23 \pm 0.62$ | $61.39 \pm 3.15$ | $80.99 \pm 0.88$ | $76.82 \pm 2.01$ | $71.49 \pm 2.74$ |
| Co-teaching+ | $78.78 \pm 0.96$ | $71.33 \pm 2.31$ | $62.90 \pm 1.55$ | $80.95 \pm 0.86$ | $77.98 \pm 0.76$ | $72.73 \pm 0.98$ |
| JoCoR | $80.11 \pm 0.42$ | $73.52 \pm 0.88$ | $63.52 \pm 0.22$ | $83.60 \pm 0.54$ | $79.45 \pm 0.38$ | $75.45 \pm 0.30$ |
| CR | $81.80 \pm 0.48$ | $74.16 \pm 0.13$ | $64.22 \pm 1.25$ | $84.09 \pm 0.39$ | $80.12 \pm 0.95$ | $75.06 \pm 1.01$ |
| NPC | $80.38 \pm 0.35$ | $72.93 \pm 0.23$ | $64.38 \pm 0.38$ | $84.28 \pm 0.47$ | $80.26 \pm 1.39$ | $73.81 \pm 0.18$ |
| LAFT (Ours) | $\mathbf{82.04 \pm 0.11}$ | $\mathbf{76.93 \pm 0.60}$ | $\mathbf{70.79 \pm 1.98}$ | $\mathbf{90.86 \pm 0.02}$ | $\mathbf{88.61 \pm 0.25}$ | $\mathbf{80.79 \pm 2.32}$ |

Table 2: The overall performance on 20Ng and AGNews with Asymmetric Noise (AN).

| Dataset | 20Ng | | AGNews | |
|---|---|---|---|---|
| AN Ratio | 20% | 40% | 20% | 40% |
| Base | 75.34 | 59.65 | 82.24 | 76.35 |
| Mixup | 75.84 | 62.19 | 82.12 | 77.45 |
| GCE | 76.41 | 62.37 | 83.19 | 77.49 |
| Co-teaching | 77.49 | 64.55 | 84.15 | 79.16 |
| Co-teaching+ | 77.98 | 63.48 | 86.12 | 81.02 |
| JoCoR | 81.27 | 70.99 | 87.63 | 82.79 |
| CR | 81.08 | 67.22 | 88.58 | 83.07 |
| NPC | 79.03 | 65.54 | 86.83 | 80.21 |
| LAFT (Ours) | **83.70** | **81.97** | **91.95** | **90.12** |

Table 3: The overall performance on 20Ng and AGNews with Instance-Dependent Noise (IDN).

| Dataset | 20Ng | | AGNews | |
|---|---|---|---|---|
| IDN Ratio | 20% | 40% | 20% | 40% |
| Base | 78.23 | 70.28 | 85.40 | 78.66 |
| Mixup | 78.21 | 70.45 | 84.71 | 79.22 |
| GCE | 79.99 | 71.74 | 87.11 | 80.49 |
| Co-teaching | 77.43 | 70.76 | 84.09 | 78.43 |
| Co-teaching+ | 79.89 | 71.52 | 88.21 | 82.30 |
| JoCoR | 81.83 | 74.04 | 88.24 | 82.66 |
| CR | 83.04 | 75.51 | 90.72 | 84.38 |
| NPC | 81.84 | 74.45 | 89.90 | 82.40 |
| LAFT (Ours) | **83.61** | **80.49** | **91.94** | **89.34** |

(3) Certain regularization methods, such as Mixup and GCE, exhibit subpar performance when applied to datasets with a higher noise ratio of 60%. This is because these methods rely on all training samples for PLM fine-tuning, making them more susceptible to strong label noises.

## 5.3 Comparison on Real-world Datasets

In this subsection, we conduct experiments on real-world datasets: SemEval (Zhou et al., 2020), TREC (Awasthi et al., 2020), and Hausa (Hedderich et al., 2020). From the results in Table 4, we observe that LAFT outperforms other baseline methods on all three datasets. Moreover, the results of LAFT are noticeably competitive on TREC and Hausa with relatively higher noise ratios, which further demonstrates the strong robustness and generalization ability of LAFT to label noise.

## 5.4 Ablation Study

In this subsection, we systematically remove specific components from our framework and ana-

lyze the resulting impact on performance. We conduct experiments with the following variants: (1) LAFT\C performs coarse-grained separation without LLMs and thus only separates the Easy Clean Set based on PLM predictions. (2) LAFT\F performs fine-grained separation without LLMs, which means when separating Hard Clean Set, only PLM-generated confidences are used. (3) LAFT\N removes our proposed loss $\mathcal{L}_{\mathcal{N}}$ for True Noisy Set and replaces it with pseudo-labeling based on the most confident PLM prediction. From the results presented in Figure 3, we observe that LAFT outperforms all variants, which verifies the effectiveness of these designs in LAFT. Specifically, removing the learning loss $\mathcal{L}_{\mathcal{N}}$ leads to significant performance degradation, indicating that such a design can effectively alleviate the adverse impact of noisy labels. Moreover, without the fine-grained separation strategy, the performance deteriorates rapidly when the noise ratio is larger, indicating the importance of fine-grained separation in the presence of higher noise ratios.

Table 4: The overall performance of various models on three real-world noisy datasets.

| Dataset | SemEval | TREC | Hausa |
|---------|---------|------|-------|
| Noise Ratio | 16.00% | 38.56% | 50.37% |
| Base | 70.61 | 67.42 | 47.80 |
| Mixup | 73.41 | 68.44 | 47.66 |
| GCE | 71.91 | 67.86 | 48.12 |
| Co-teaching | 72.13 | 68.19 | 47.56 |
| Co-teaching+ | 72.24 | 69.83 | 48.25 |
| JoCoR | 70.11 | 66.62 | 46.48 |
| CR | 72.94 | 68.90 | 48.34 |
| NPC | 71.22 | 67.84 | 47.48 |
| LAFT (Ours) | **73.56** | **72.34** | **51.71** |

Table 5: The LLM-generated prediction accuracy on the training set in 20Ng and AGNews in the *ideal* separation and real separation (shown in gray) during fine-tuning.

| Dataset | EC Set | HC Set | TN Set | Overall |
|---------|--------|--------|--------|---------|
| 20Ng | 99.46% | 0% | 76.47% | 75.34% |
| | 99.46% | 3.58% | 69.44% | 75.34% |
| AGNews | 99.03% | 0% | 79.35% | 81.41% |
| | 99.03% | 2.05% | 76.07% | 81.41% |

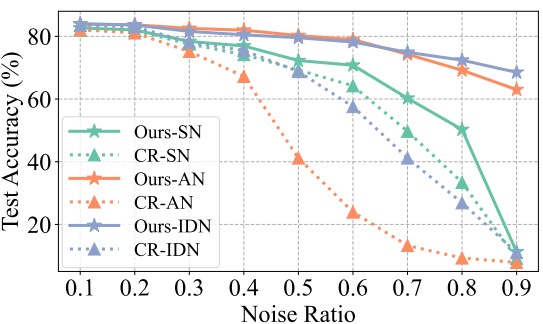

Figure 4: The results of our framework and the best baseline CR on 20Ng with different noise ratios for three types of noise: SN, AN, and IDN.

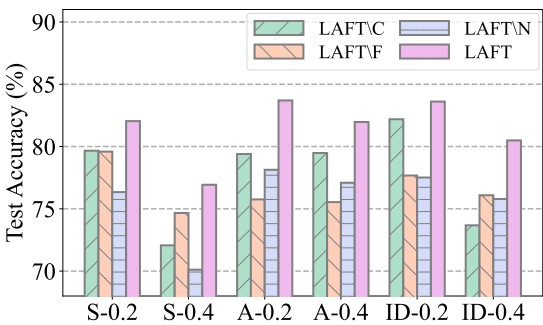

Figure 3: Ablation study of our framework on 20Ng. S-0.2 and S-0.4 (A, or ID) refer to the noise ratios of 20% and 40% for SN (AN, or IDN), respectively.

## 5.5 Evaluation of LLMs-generated Labels

In this subsection, we evaluate the effectiveness of LLMs regarding the quality of generated confidence, presented in Table 5. Specifically, we report the accuracy of LLM-generated prediction on the training set in 20Ng and AGNews. The statistics are from experiments on 20Ng and AGNews with 20% Symmetric Noise (SN). From the results, we can observe that: (1) For Easy Clean Set, the LLM-generated labels exhibit a remarkably high degree of correctness. This empirical finding provides the justification for Assumption 1, which allows for leveraging LLMs to perform coarse-grained separation in our framework. (2) For the TN set, LLM-generated labels are not entirely correct, while still preserving similar accuracy compared to that on all samples. This result empirically verifies Remark 1 and justifies our strategy of utilizing LLM-generated confidences for learning from the TN set. (3) Considering the overall result, LLM-generated labels generally exhibit lower accuracy than PLM-based baselines. That being said, LLMs cannot be directly used to replace PLMs for the text classification task when the noise ratio is not extremely high. Nevertheless, our strategy can effectively leverage the LLM-generated confidences, despite their lack of complete correctness, to enhance the fine-tuning performance of PLMs with noisy labels.

## 5.6 Results with Different Noise Ratios

Figure 4 presents the evaluation of our proposed framework across a spectrum of noise ratios on 20Ng, ranging from 0.1 to 0.9, while considering three distinct types of noise: SN, AN, and IDN. From the results, several significant observations are discovered: (1) With increasing noise ratios, all baseline methods consistently show a deterioration in performance. This decline can be tied to the amplified difficulty of label noise during PLM fine-tuning. (2) Despite the escalating noise ratios, our framework maintains a relatively more robust performance. This is attributed to our sample separation strategy which adeptly discerns clean samples within the training set, thus alleviating the adverse effects of label noise. (3) Our framework demonstrates a slower performance decrease with AN noise compared to other types of noise. This can be credited to the adaptive nature of our approach that employs LLM-generated confidences to effectively learn from similar labels for each sample.

# 6 Conclusion

This paper delves into the issue of fine-tuning Pretrained Language Models (PLMs) with noisy labels for text classification tasks. We address the detrimental effects of label noise by harnessing external guidance from Large Language Models (LLMs) in the form of confidences. Our approach entails a two-step separation strategy that accurately segregates all training samples into three distinct subsets, each treated with innovative noise-resilient strategies. In summary, our framework adeptly fine-tunes PLMs in the face of noisy samples, requiring minimal external guidance from LLMs.

## Acknowledgements

Song Wang and Jundong Li are supported by the National Science Foundation under grants (IIS-2006844, IIS-2144209, IIS-2223769, CNS2154962, and BCS-2228534), the Commonwealth Cyber Initiative awards (VV-1Q23-007, HV-2Q23-003, and VV-1Q24-011), the JP Morgan Chase Faculty Research Award, the Cisco Faculty Research Award, the Jefferson Lab subcontract, and the UVA 4-VA collaborative research grant.

## Limitation

Despite the promising results, our work presents several limitations that we must acknowledge: (1) Dependence on Large Language Models (LLMs): Our framework leverages the guidance from LLMs in the form of confidences. This implies that the quality of these confidences and hence the effectiveness of our model are closely tied to the performance of the LLM. Unsatisfactory LLM performance could thus directly impact our framework's efficacy. (2) Noise Types and Ratios: The experiments in this paper mainly focus on three types of noise: Symmetric Noise (SN), Asymmetric Noise (AN), and Instance-Dependent Noise (IDN), with noise ratios up to 60%. However, there may be other noise types or higher noise ratios in practice, and it remains to be investigated how well our framework performs under these circumstances. (3) Limitations in Sample Separation Strategies: The efficacy of our framework relies on accurately dividing the training samples into clean and noisy subsets. If this distinction is not clear or becomes more complex, our current approach may encounter difficulties. (4) Domain-Specific Text Classification: While the experiments are performed on specific text classification tasks, we do not investigate the effectiveness of our approach in more domain-specific contexts where the nature of the noise could be different. (5) Computational Costs: Finally, our approach entails using LLMs which can be computationally expensive and could thus pose a challenge when applied to large datasets or in resource-constrained environments. In summary, future research could focus on overcoming these limitations and exploring the adaptability of our proposed framework to other noise types, higher noise ratios, more complex noise patterns, as well as different task domains.

## Ethics Statement

This research adheres to the principles of ethical conduct in artificial intelligence research. The development and utilization of our work are carried out with full recognition of the potential implications. While our method improves the performance of Pretrained Language Models (PLMs) with noisy labels in text classification tasks, we acknowledge that it could be potentially used in contexts that may involve ethical concerns, such as disinformation or manipulation of opinion on social platforms. Our experiments are performed on five publicly available datasets, namely 20Ng, AGNews, SemEval, TREC, and Hausa. The datasets are already anonymized and do not contain any personally identifiable information. Moreover, we have complied with all guidelines and standards for the use of these datasets. Furthermore, the use of Large Language Models (LLMs) in our work is subject to the terms and conditions put forth by the creators and hosts of these models. We only emply them for the purpose of enhancing the fine-tuning performance of PLMs and do not exploit them for any inappropriate or unauthorized purposes. While our work focuses on improving the robustness of PLMs to label noise in data, we recognize the broader societal concerns related to the potential misuse of our framework. We advocate for the responsible use of our research findings and encourage continued discourse around the ethical use of machine learning and artificial intelligence technologies in real-world applications. Finally, it is crucial to note that the development of technologies and methodologies in machine learning should be done in conjunction with ongoing considerations about their ethical implications, potential misuse, and societal impacts. It is our responsibility to foster an environment of ethical vigilance in AI research and development.

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

# A  Details of Three Subsets

In this section, we further provide details about the three subsets of samples in our framework.

1. **Easy Clean (EC) Set.** In this subset, the labels predicted by LLMs are the same as the assigned labels. Therefore, based on Assumption 1, we can infer that the labels of these samples are almost clean, which allows us to directly use them for model training. Note that a small portion of samples in this category can still be noisy when the LLM predictions happen to be the same as noisy labels. However, we empirically verify in Sec. 5.5 that this proportion is small.

2. **Hard Clean (HC) Set.** In this (ideal) category, the labels predicted by LLMs deviate from the assigned labels, while the true labels are in accordance with the assigned labels. In this case, we are acknowledged that these samples are also clean. However, due to the semantic difficulties of these samples, the LLMs cannot easily predict the correct labels of them. Nevertheless, since the assigned labels are correct as true labels, we can use assigned labels for model training on these samples. It is noteworthy that in practice, it is infeasible to achieve a perfect separation of this subset. Therefore, in our framework, we propose to leverage LLM-generated and PLM-generated confidences to improve the separation performance.

3. **True Noisy (TN) Set.** In this (ideal) category, the labels predicted by LLMs and the true labels are both different from the assigned labels. In other words, these samples are all noisy as their assigned labels are different from true labels. Since they are noisy, the labels predicted by LLMs are also likely to be different from the assigned labels. As these samples are noisy, we cannot directly use their labels for training. However, since we narrow down the potential range of noisy samples to this category, we can resort to specific techniques to learn from these noisy samples. In our framework, we utilize LLM-generated confidences as additional supervision information to effectively learn from these noisy samples.

## B  Datasets

In this section, we introduce the details of the datasets used in our experiments. In particular, 20Ng (Lang, 1995) and AGNews (Li and Roth, 2002; Zhang et al., 2015) are news topic classification datasets that are prevalently used for text classification tasks. We manually inject different types of noise (SN, AN, and IDN) into these two datasets for evaluation in our experiments. For real-world datasets, we utilize SemEval (Zhou et al., 2020), TREC (Awasthi et al., 2020), and Hausa (Hedderich et al., 2020). Specifically, SemEval is a relation extraction dataset, and we follow the process introduced in (Zhou et al., 2020) to obtain noisy labels. TREC is a question classification dataset in the weak supervision benchmark WRENCH (Zhang et al., 2021b). Moreover, Hausa is a text classification dataset in the language of

Table 6: The detailed statistics of the five datasets used in our experiments.

| Dataset | # Training | # Validation | # Test | # Class |
|---------|-----------|-------------|--------|---------|
| 20Ng    | 9,051     | 2,263       | 7,532  | 20      |
| AGNews  | 40,000    | 7,600       | 7,600  | 4       |
| SemEval | 1,749     | 200         | 692    | 9       |
| TREC    | 4,965     | 500         | 500    | 6       |
| Hausa   | 2,045     | 290         | 582    | 5       |

Hausa, the second most spoken indigenous language in Africa, with 40 million native speakers. For this dataset, gazetteers are used for automatic labeling, which results in feature-dependent label noise. Here we provide the detailed statistics of these datasets in Table 6.

## C  Baselines

In this section, we provide the details of the baselines used in our experiments.

- Base (Devlin et al., 2018) is the BERT base model fine-tuned based on the standard cross-entropy loss. Note that for dataset Hausa, we utilize the multilingual BERT model.

- Mixup (Zhang et al., 2018) is a semi-supervised approach that performs a linear interpolation between clean and noisy samples.

- GCE (Zhang and Sabuncu, 2018) denotes Generalized Cross-Entropy loss, which can be regarded as a general loss that combines mean absolute error (MAE) and cross-entropy (CE).

- Co-teaching (Han et al., 2018) trains two different models and selects small-loss samples to feed them into each other for optimization.

- Co-teaching+ (Yu et al., 2019) is an updated version of Co-teaching that explicitly ensures the difference between the two models.

- JoCoR (Wei et al., 2020) trains two models and selects samples based on the sum of loss from the two models.

- CR (Zhou and Chen, 2021) also trains multiple models with a regularization strategy based on a soft target.

- NPC (Bae et al., 2022) utilizes a generative model to estimate the transition matrix from noisy predictions to the ground-truth labels of samples and uses it to correct noisy labels.

## D  Implementation Details

In this section, we introduce the implementation details for our experiments. We run the experiments on a single 48GB Nvidia GeForce RTX A6000 GPU. The experiments are run 10 times to achieve the average accuracy and the standard deviation.

For the augmentations used in our framework, following existing works (Xie et al., 2019; Wei and Zou, 2019; Li et al., 2022), we adopt four typical text augmentations: Back Translation, Random Insertion, Random Deletion, and Random Swap. We set the threshold $\hat{\tau}$ for LLM-generated confidences as 0.8. We set the hyper-parameters $\lambda$ and $\widetilde{\tau}$ for the adaptive threshold $\widetilde{\tau}(t)$ for PLM-generated confidences as 0.7 and 0.8, respectively. We set $\alpha$ as 1.5 and $\beta$ as 0.9. For the loss weight, we set $\lambda_{\mathcal{H}} = \lambda_{\mathcal{N}} = 1$. We set the model fine-tuning learning rate as $10^{-4}$ with a batch size of 32. The maximum length of texts is set as 256. The number of training epochs is set as 100 with early stopping based on the performance of the validation set. We optimize the model using the Adam (Kingma and Ba, 2015) optimization strategy. For the LLM, we utilize ChatGPT based on GPT4.

In addition, we provide more detailed package requirements as listed below.

- Python == 3.9.12
- torch == 2.0.1
- transformers == 4.29.2
- huggingface-hub == 0.15.1
- keras == 2.12.0
- numpy == 1.23.5
- scipy == 1.5.3
- cuda == 11.6
- networkx == 2.5.1
- scikit-learn == 0.24.1