# OpenReview forum: "Noise-Robust Fine-Tuning of Pretrained Language Models via External Guidance"
_EMNLP/2023/Conference — EMNLP 2023 Findings_

### Official Review · Reviewer_Qwqa · 2023-08-01

**Soundness:** 4

**Excitement:**

3: Ambivalent: It has merits (e.g., it reports state-of-the-art results, the idea is nice), but there are key weaknesses (e.g., it describes incremental work), and it can significantly benefit from another round of revision. However, I won't object to accepting it if my co-reviewers champion it.

**Missing References:**

Is noise robust equals to cleaning all the noise and training with clean data? Some discussion is here: https://arxiv.org/pdf/2202.12024.pdf

**Paper Topic And Main Contributions:**

The paper focus on the study of fine-tuning PLM with noisy labels, where they provide a method to use LLM as external guidance to filter the clean data and noisy data.

**Questions For The Authors:**

1. In Section 4.3, from line 301, the author directly jumped from the motivations to the detailed methods. Why is the threshold set for the highest value of the max confidence score? Is that based on the intuition that LLMs perform well in identifying true noisy labels?

2. What is the overall efficiency of the methods of separating clean samples from the training dataset, compared with other methods in noise-robustness?

3. In section 4.5, is it possible to use negative sampling as a way to process a true noisy set?

4. When using a more powerful model to filter the data for a smaller model like LLMs to filter clean data for BERT-based PLM. What do you think the best performance of the PLM will be (close to the LLM or even better, or others) and why? Even if not fine-tuning LLMs, do you consider it a fair comparison between your approach and other PLM-based approach, when adopting LLM as a filter?

**Reasons To Accept:**

Overall, the method is written concisely and clearly and is easy to read and understand.

The methods of incorporating LLMs as a tool for the benefits of PLM are also a popular and promising direction.

The user uses experiments and data to support their motivation of these challenges, and propose solutions accordingly, which is effective and accurate.

**Reasons To Reject:**

The study is based on the intuitive approach of separating clean samples from the training set for model optimization can benefit the PLM. From a large perspective, there is no discussion on the efficiency or benefits of this approach compared with other approaches for noisy labels. The user also fails to extend his idea to other areas in NLP with noisy data labelling or other LLM-based models.

Even though the methods with LLMs are proposed, the LLMs play only the role of a more powerful classification model, which can also be replaced with other methods. There lacks an in-depth discussion on the motivation for using LLMs and how it benefits the overall process.

**Reproducibility:**

4: Could mostly reproduce the results, but there may be some variation because of sample variance or minor variations in their interpretation of the protocol or method.

**Reviewer Confidence:**

4: Quite sure. I tried to check the important points carefully. It's unlikely, though conceivable, that I missed something that should affect my ratings.

**Typos Grammar Style And Presentation Improvements:**

Figure 4 is not mentioned in the paragraph of section 5.4. It can be analysed more, for example, why LAF\C is dropping with noise ratio increasing?

---

> ### Author Rebuttal · Authors · 2023-08-29
>
> Dear Reviewer Qwqa:
>
> Thanks for your insightful feedback and suggestions. Here we would like to address your questions as follows:
>
>   &nbsp;
>
>
> > W1: The study is based on the intuitive approach of separating clean samples from the training set for model optimization can benefit the PLM. From a large perspective, there is no discussion on the efficiency or benefits of this approach compared with other approaches for noisy labels. The user also fails to extend his idea to other areas in NLP with noisy data labeling or other LLM-based models
>
>
> > Q2: What is the overall efficiency of the methods of separating clean samples from the training dataset, compared with other methods in noise-robustness?
>
>
>
>
> A1: Thank you for highlighting the intuition and potential comparisons of our framework. We apologize for not comparing the efficiency of our framework with other baselines. We will include a comprehensive discussion in our Limitation section.
>
>
>
> We would also like to briefly discuss the efficiency of our framework here. The main time consumption results from the process of querying an LLM for each sample. Nevertheless, we observe that with the guidance of LLMs, the fine-tuning of PLMs can be more efficient. Moreover, with the separated data, our framework only trains **one** network, in contrast to prevalent baselines that train **multiple** networks (Co-teaching, Co-teaching+, JoCoR, CR, and NPC). The guidance from the LLMs ensures a **faster convergence** for fine-tuning PLMs, as shown in the results below.
>
>
>
> Here we include the results of running time until convergence on 20Ng and AGNews (20% SN) as follows, trained on a single 48GB Nvidia GeForce RTX A6000 GPU.
>
>
>
> |Model| 20Ng| AGNews|
> |:----------:|:-------------:|------:|
> |Co-teaching| 431.21s| 993.48s|
> |Co-teaching+| 598.63s| 1706.1s|
> |JoCoR| 1078.39s| 2357.55s|
> |CR| 1380.39s| 3510.82s|
> |NPC| 304.18s| 1068.11s|
> |LAFT (Ours)| **265.09s**| **768.19s**|
>
>
>
>
> From the results, we observe that our framework **surpasses other baselines** in terms of running time. This is primarily due to that we only fine-tune one network instead of multiple ones in baselines. It is noteworthy that another time consumption is about querying the LLMs. Nevertheless, as we utilize black box LLMs, it is feasible to pre-process the queries via provided LLM APIs (nearly 1 second for each sample via ChatGPT). Such a process does not require computational resources and can be achieved in parallel with fine-tuning.
>
>
>
>
> Regarding the extension of our idea to other areas, thanks for your valuable suggestion! While our work mainly focuses on text classification, which is a fundamental and crucial problem in NLP, we recognize the potential of our method in various NLP tasks with noisy data. In future work, we will also consider applying our idea in other NLP tasks as well as incorporating it into other methods based on LLMs. To precisely summarize our work, we would like to revise our title to “Noise-Robust Fine-Tuning of Pretrained Language Models via External Guidance for Text Classification”.
>
> Regarding generalizability, as our framework utilizes **black-box** LLMs, we believe it can be easily generalized to a variety of LLM-based methods, e.g., in-context learning and reasoning.
>
>
>
>   &nbsp;
>
>
> > W2: Even though the methods with LLMs are proposed, the LLMs play only the role of a more powerful classification model, which can also be replaced with other methods. There lacks an in-depth discussion on the motivation for using LLMs and how it benefits the overall process.
>
>
>
> A2: Thank you for highlighting the role of LLMs in our work. We would like to clarify that the significance of our work lies in the proposed methodology to separate clean and noisy data and also learn from them under external guidance from LLMs. We leverage LLMs as external guidance because they bear prior knowledge that can provide useful information in both separation and learning (with our designed loss).
>
> We agree that other models (e.g., PLMs) can also generate confidences. However, as illustrated in our Sec. 4.1, they tend to output high confidences for noisy samples and thus will harm the performance, due to the incorrect separation. That being said, the model will treat potentially noisy samples as clean data and thus learn incorrect information. To verify the superiortiy of using LLMs instead of PLMs, we conducted an additional set of experiments, where **the confidences** are generated by various PLMs using the classification probabilities. We report the results on datasets 20Ng and ANNews with 20% Symmetric Noise (SN).
>
>
>
> |Model (Accuracy)| 20Ng| AGNews|
> |:----------:|:-------------:|:-------------:|
> |LAFT w/ BERT| 78.60%| 82.31%|
> |LAFT w/ RoBERTa| 80.05%| 85.09%|
> |LAFT w/ GPT-2| 77.92%| 84.17%|
> |LAFT w/ ChatGPT (Ours)|**82.04%**|**90.86%**|
>
>
> From the results, we can observe that using LLMs can outperform the methods with PLMs. This is because the LLMs provide more precise confidences and are also more robust to noise. We appreciate your feedback and will ensure that our revised version offers a more comprehensive understanding of the motivation and benefits of incorporating LLMs.
>
>
>
>   &nbsp;
>
>
> > Q1: In Section 4.3, from line 301, the author directly jumped from the motivations to the detailed methods. Why is the threshold set for the highest value of the max confidence score? Is that based on the intuition that LLMs perform well in identifying true noisy labels?
>
>
>
>
> A1: Thank you for pointing out the confusion. We set the threshold for the highest confidence for two reasons:
>
>
>
> 1. The strategy of selecting samples based on the highest confidence is **widely adopted** by existing methods for learning from noisy labels [1,2,3]. This is because the highest confidence reflects the model’s certainty on a specific sample.
> A high confidence usually indicates that a sample can be easily classified, suggesting that it is more likely to be clean. Such an approach is not only intuitive but is also empirically validated as effective in multiple works.
>
>
> 2. The highest confidence is **consistent** across datasets. Given that the confidences for all classes of a sample sum up to 1, the range of the highest confidence value is bound in $[1/N,1]$, where $N$ is the number of classes. This consistent behavior makes the threshold more robust and generalizable to various datasets.
>
>
>
> [1] Zhang, Bowen, et al. "Flexmatch: Boosting semi-supervised learning with curriculum pseudo labeling." In NeurIPS 2021.
> [2] Xie, Qizhe, et al. "Unsupervised data augmentation for consistency training." In NeurIPS 2020.
> [3] Nishi, Kento, et al. "Augmentation strategies for learning with noisy labels." In CVPR 2021.
>
>
>
>
>
>
>
>   &nbsp;
>
>
>
>
> > Q3: In section 4.5, is it possible to use negative sampling as a way to process a true noisy set?
>
>
>
> A3: We appreciate your constructive suggestion regarding the potential use of negative sampling. It is indeed an intriguing idea to leverage noisy data. In our framework, samples in the TN set are mostly noisy, making them potential candidates for negative samples in contrastive learning. However, such a strategy **cannot fully utilize the information** provided by LLMs, because it cannot utilize the LLM-generated confidences in other classes. In contrast, our designed loss in Sec. 4.5 can help PLMs learn from other potentially correct labels based on LLM-generated confidences. We value your feedback and will incorporate this discussion in our revised version.
>
>
>
>
>   &nbsp;
>
>
> > Q4: When using a more powerful model to filter the data for a smaller model like LLMs to filter clean data for BERT-based PLM. What do you think the best performance of the PLM will be (close to the LLM or even better, or others) and why? Even if not fine-tuning LLMs, do you consider it a fair comparison between your approach and other PLM-based approach, when adopting LLM as a filter?
>
>
>
>
> A4: Thank you for your thoughtful query regarding the comparison between our framework and other PLM-based baselines.
>
>
> We believe (and also observe) that the performance of PLMs fine-tuned on LLM-filtered data can surpass the performance of **solely using LLMs** (Table 5). For example, LLMs achieve 75.34% on 20Ng (irrelevant to noise ratios as LLMs are not fine-tuned), while our method can achieve 76.93% even with 40% noisy data for fine-tuning. The results demonstrate that the filtered data can significantly promote the fine-tuning performance of PLMs.
>
>
>
>
> In terms of the comparison between our framework and other PLM-based methods, we believe it is justifiable. This is because we do not directly transfer knowledge, but use the LLM's predictions to guide the training of the PLM. Such a strategy enables us to leverage **black-box LLMs** for fine-tuning PLMs without significant computational costs. Moreover, our framework does not need to use LLMs during inference, which means **the practicality** is the same as other PLM-based methods.

---

### Official Review · Reviewer_No3t · 2023-08-05

**Typos Grammar Style And Presentation Improvements:** superior advantages -> advantages
**Soundness:** 2

**Excitement:**

3: Ambivalent: It has merits (e.g., it reports state-of-the-art results, the idea is nice), but there are key weaknesses (e.g., it describes incremental work), and it can significantly benefit from another round of revision. However, I won't object to accepting it if my co-reviewers champion it.

**Missing References:**

One important reference that is clearly relevant is label smoothing, as introduced in:
@misc{szegedy2015rethinking,
      title={Rethinking the Inception Architecture for Computer Vision},
      author={Christian Szegedy and Vincent Vanhoucke and Sergey Ioffe and Jonathon Shlens and Zbigniew Wojna},
      year={2015},
      eprint={1512.00567},
      archivePrefix={arXiv},
      primaryClass={cs.CV}
}

and studied in:

@misc{müller2020does,
      title={When Does Label Smoothing Help?},
      author={Rafael Müller and Simon Kornblith and Geoffrey Hinton},
      year={2020},
      eprint={1906.02629},
      archivePrefix={arXiv},
      primaryClass={cs.LG}
}

**Paper Topic And Main Contributions:**

The paper tackles a relevant problem, namely mitigating the effects of noisy training data for fine-tuning pre-trained language models.

It proposes to do so by employing an external large language model.


**Reasons To Accept:**

Mitigating noise in data is of general interest and any significant contribution is likely to have wide impact.


**Reasons To Reject:**

A basic concept throughout the paper is undefined: confidence of both pre-trained model and external LLM are undefined.

There is also the possibility of information leaking from the LLM into the pre-trained LM, making it unclear to what extent the approach proposed is not in fact a form of distillation.

Usually fine-tuning data is scarce and critical for good performance on the domain/task of interest, so making sure it is of high quality (not noisy) is a top priority.

**Reproducibility:**

3: Could reproduce the results with some difficulty. The settings of parameters are underspecified or subjectively determined; the training/evaluation data are not widely available.

**Reviewer Confidence:**

4: Quite sure. I tried to check the important points carefully. It's unlikely, though conceivable, that I missed something that should affect my ratings.

---

> ### Author Rebuttal · Authors · 2023-08-28
>
> Dear Reviewer No3t:
>
> Thanks for your valuable feedback and suggestions on the missing reference! We would like to address your questions as follows:
>
>   &nbsp;
>
> > **W1: A basic concept throughout the paper is undefined: confidence of both pre-trained model and external LLM are undefined.**
>
> A1: Thank you for highlighting your concern regarding the definition of confidence for both pre-trained language models (PLM) and LLMs. We wish to clarify that we have **indeed defined** the confidence of PLMs on **Line 219-224** in **Eq. (1)**, and the confidence of LLMs on **Line 243-245** in our paper. We really appreciate your feedback and acknowledge that the presentation could be clearer. To address this issue, we will insert a formal definition of "confidence" in the revised version to enhance clarity. We value your suggestion and will ensure that the concept is presented more clearly in our revision. The following definition is obtained based on Line 219-224.
>
>
> ***Definition**. The confidence score denotes the certainty when a model classifies a sample, regarding a specific class. Generally, the confidences are represented by the classification probabilities. Denote the output of the final layer (i.e., the classifier) in PLMs for sample $x_i$ as $\mathbf{z}(x_i)\in\mathbb{R}^{N}$, where $N$ is the number of classes.
> The confidence of $x_i$ for the $j$-th class $c_j$ can be represented as follows:*
> $$
> \widetilde{p}(c_j; x_i)=\frac{\exp\left(z(c_j; x_i)\right)}{\sum_{k=1}^N \exp\left(z(c_k; x_i)\right)},
> $$
> *where $z(c_j;x_i)\in\mathbb{R}$ is the $j$-th value in $\mathbf{z}(x_i)$. Notably, the confidences are obtained from $\mathbf{z}(x_i)$ after a softmax function and thus sum up to one.*
>
>   &nbsp;
>
> > **W2: There is also the possibility of information leaking from the LLM into the pre-trained LM, making it unclear to what extent the approach proposed is not in fact a form of distillation.**
>
> A2: Thank you for raising the concern about potential information leakage and the similarity between our framework and knowledge distillation.
>
> Although our work is related to knowledge distillation, we wish to clarify that our framework **differs from** traditional knowledge distillation from two unique perspectives.
>
>
> 1. Rather than having the PLMs directly mimic the labels and probabilities produced by LLMs [1,2,3], we utilize LLMs as **external guidance** for fine-tuning PLMs. This guidance, in the form of predictions and confidences from LLMs, aids the **sample separation** process in our pipeline. Additionally, this information from LLMs is also crucial in enabling PLMs to effectively learn from noisy data.
>
> 2. In contrast to knowledge distillation, in our framework, PLMs fine-tuned under the guidance of LLMs can outperform solely using LLMs (demonstrated in Sec. 5.5). The benefits come from the sample selection process, instead of directly learning knowledge from LLMs.
>
>
>
> To better illustrate the difference between our work and knowledge distillation, we conduct experiments using traditional knowledge distillation based on predictions and confidences from LLMs. The results in accuracy are presented as follows (20% Symmetric Noise for 20Ng and AGNews):
>
> |Model (Accuracy)| 20Ng| AGNews|SemEval|TREC|Hausa|
> |:----------:|:-------------:|:-------------:|:-------------:|:-------------:|:------:|
> |Base|79.15%|81.77%|70.61%|67.42%|47.8%|
> |ChatGPT| 75.34%| 81.41%| 63.35%|68.76%| 50.17%|
> |KD w/ ChatGPT Confidences| 79.33%| 85.61%| 67.68%| 68.94%| 48.06%|
> |KD w/ ChatGPT Predictions| 77.93%| 83.32%| 67.09%| 67.11%| 46.80%|
> |LAFT (Ours)|**82.04%**|**90.86%**| **73.56%**| **72.34%**|**51.71%**|
>
>
>
> From the results, we can observe that directly distilling knowledge from LLMs **cannot outperform our framework**. Moreover, when the performance of LLM is inferior (e.g., on SemEval), the KD methods are even less competitive than the Base method.
>
>
>
> Since LLMs are not fine-tuned on this specific dataset, their prior knowledge may not provide useful information regarding the classification. Nevertheless, in our framework, we can effectively leverage the confidence produced by LLMs to separate clean and noisy data for fine-tuning PLMs. With our designed loss to learn from separated data, our framework can achieve outperforming results.
>
>
>
> [1] Magister, Lucie Charlotte, et al. "Teaching small language models to reason." arXiv preprint arXiv:2212.08410 (2022).
> [2] Fu, Yao, et al. "Specializing Smaller Language Models towards Multi-Step Reasoning." arXiv preprint arXiv:2301.12726 (2023).
> [3] Hsieh, Cheng-Yu, et al. "Distilling step-by-step! outperforming larger language models with less training data and smaller model sizes." arXiv preprint arXiv:2305.02301 (2023).
>
>
>   &nbsp;
>
>
> > **W3: Usually fine-tuning data is scarce and critical for good performance on the domain/task of interest, so making sure it is of high quality (not noisy) is a top priority.**
>
>
>
> A3: Thanks for your insightful opinion. We agree on the importance of clean data used for fine-tuning the pre-trained language models (PLMs), especially in scenarios where domain knowledge is necessary. However, for many downstream tasks, the labels can be noisy due to the labor-intensive and time-consuming labeling process [1,2].
>
> In fact, this is **the motivation** of our framework for learning from noisy labels. We aim to maximally leverage information from both clean and noisy data, and thus propose to **separate clean data** under the guidance of LLMs.
>
>
>
> To verify that our framework can indeed separate clean data, we present the percentages of the Easy Clean (EC) set and the LLM accuracy on it. We show the detailed statistics on datasets 20Ng and AGNews with 20% Symmetric Noise as follows:
>
> The **LLM accuracy** on each set:
> | Accuracy| Easy Clean (EC) | Hard Clean (HC)| True Noisy (TN)| Overall|
> |:----------:|:-------------:|:-------------:|:------:|:------:|
> | 20Ng| 99.46%|3.58%| 69.44% | 75.34%|
> |AGNews|99.03% |2.05%| 76.07% |81.41%|
>
> The **percentage** of each set:
> | Percentage|  Easy Clean (EC) | Hard Clean (HC)| True Noisy (TN)|  Overall|
> |:----------:|:-------------:|:-------------:|:------:|:------:|
> | 20Ng| 60.91% (5,513) | 18.80% (1,702) | 20.29% (1,836) | 100% (9,051)|
> | AGNews| 65.31% (26,124) | 13.04% (5,217) | 21.64% (8,659) | 100% (40,000)|
>
>
>
> From the results we can observe that, our framework can effectively separate an Easy Clean set with relatively sufficient data (larger than 60% of all data). The LLM accuracy is also high on this set, indicating that the data is **mostly clean**.
>
> To evaluate the **quality of the separated EC set**, we additionally report the result when fine-tuning PLMs with only the EC set.
> |Method (Accuracy)| 20Ng| AGNews|
> |:----------:|:-------------:|:------:|
> |ChatGPT| 75.34%|81.41%|
> |Fine-tune w/ All (Base) |79.15%|81.77%|
> |Fine-tune w/ EC |81.04%|83.49%|
> |LAFT (Ours) |**82.04%**|**90.86%**|
>
> We can observe that the performance of fine-tuning with only the EC set **surpasses the Base method** and the LLM. The results demonstrate that the EC set separated by our framework indeed **contains useful and important information** for classification. Nevertheless, only using the EC set cannot outperform our framework LAFT, which demonstrates the importance of also learning from noisy data with our strategy.
>
> [1] Zhou, Wenxuan, and Muhao Chen. "Learning from Noisy Labels for Entity-Centric Information Extraction." In EMNLP 2021.
> [2] Wei, Hongxin, et al. "Combating noisy labels by agreement: A joint training method with co-regularization." In CVPR 2020.
>
>   &nbsp;
>
>
> > **W4: One important reference that is clearly relevant is label smoothing.**
>
>
>
>
> A4: Thank you for your valuable suggestion regarding the significant reference to label smoothing. We will incorporate the relevant works of label smoothing [1,2,3,4] in the revised version of our paper.
>
>
>
> We would also like to clarify the difference between our framework and label smoothing from two perspectives.
>
> 1.  **Objectives**. Label smoothing aims to deal with overfitting and lack of generalizability. In contrast, our work proposes to **separate** a clean set from all noisy data for learning from noisy labels.
>
> 2.  **Strategies**. Label smoothing is achieved by regularizing the classification probability. However, our work harnesses the predictions and confidences provided by **LLMs** to effectively separate clean data and also help fine-tune PLMs with noisy data.
>
>
>
>
> [1] Szegedy, Christian, et al. "Rethinking the inception architecture for computer vision." In CVPR 2016.
> [2] Müller, Rafael, Simon Kornblith, and Geoffrey E. Hinton. "When does label smoothing help?." In NeurIPS 2019.
> [3] Lukasik, Michal, et al. "Does label smoothing mitigate label noise?." In ICML 2020.
> [4] Wei, Jiaheng, et al. "To Smooth or Not? When Label Smoothing Meets Noisy Labels." In ICML 2022.

---

### Official Review · Reviewer_br2H · 2023-08-05

**Soundness:** 3

**Excitement:**

4: Strong: This paper deepens the understanding of some phenomenon or lowers the barriers to an existing research direction.

**Paper Topic And Main Contributions:**

To address the noise label problem in text classification, this paper proposes LAFT, a pre-trained language model fine-tuning method with guidance from larger language models. LAFT separates the training examples into three parts (easy clean, hard clean, and true noisy) and utilizes three objectives to learn from different categories of examples. This paper conducts exhaustive experiments on synthetic and real-world noisy datasets to demonstrate the effectiveness of LAFT.

**Questions For The Authors:**

A: In Figure 4, LAFT with 0.9 ratio noise from AN or IDN obtains nearly 60% accuracy. Why the accuracy under these settings do not have a significant degradation?

B: What happens when the noise ratio becomes 0, but the framework doesn't know this prior ratio while keeping the assumption that there are noises in the dataset?

**Reasons To Accept:**

1. The idea of utilizing LLMs to help improve PLMs fine-tuning is worth exploring.
2. This paper is well-written and can be easy to read.
3. The work is solid, and the experiments are exhaustive. Details are provided so that this work can be followed easily.

**Reasons To Reject:**

1. As shown in the Limitation section, the performance of LAFT is highly correlated to the quality of the LLMs, which limits the framework's potential.
2. The assumption, as well as the remark, are too strong. 20Ng and AGNews are well-known public datasets that can be regarded as "very clean" datasets. Therefore, the LLM's accuracy in the EC (easy clean) category should be intuitively high. The accuracy in the EC category under the truly-noised dataset will be more convincing. Meanwhile, the percentage of the three categories (EC, HC, and TN) is missed.

**Reproducibility:**

5: Could easily reproduce the results.

**Reviewer Confidence:**

4: Quite sure. I tried to check the important points carefully. It's unlikely, though conceivable, that I missed something that should affect my ratings.

---

> ### Author Rebuttal · Authors · 2023-08-28
>
> Dear Reviewer br2H,
>
> Thanks for your insightful feedback and suggestions. Here we would like to address your questions as follows:
>
> &nbsp;
> > W1: As shown in the Limitation section, the performance of LAFT is highly correlated to the quality of the LLMs, which limits the framework's potential.
>
> A1: Thank you for emphasizing the relationship between the performance of LAFT and the quality of LLMs, as highlighted in our Limitation section. While we acknowledge this dependence, we also view it as a **potential strength**. As our framework LAFT is designed to flexibly leverage various (black box) LLMs, higher-quality LLMs can provide more comprehensive knowledge and thus further promote performance, thus an advantage of our framework. There also exists a considerable number of works that harness the power of LLMs on specific tasks for smaller models [1,2,3].
>
> [1] Tamkin, Alex, et al. "Active learning helps pretrained models learn the intended task." In NeurIPS 2022.
> [2] Bansal, Parikshit, and Amit Sharma. "Large Language Models as Annotators: Enhancing Generalization of NLP Models at Minimal Cost." arXiv preprint arXiv:2306.15766 (2023).
> [3] Thirunavukarasu, Arun James, et al. "Large language models in medicine." Nature Medicine (2023).
>
>
>
>
> &nbsp;
>
>
> > W2: The assumption, as well as the remark, are too strong. 20Ng and AGNews are well-known public datasets that can be regarded as "very clean" datasets. Therefore, the LLM's accuracy in the EC (easy clean) category should be intuitively high. The accuracy in the EC category under the truly-noised dataset will be more convincing. Meanwhile, the percentage of the three categories (EC, HC, and TN) is missed.
>
> A2: Thank you for highlighting your concerns regarding our assumptions and the clarity of our remarks on the datasets. We believe that the assumption is achievable, as the LLM’s accuracy in the EC (easy clean) set is **not directly related** to the difficulty of a dataset. Particularly, for each sample misclassified by the LLM within the EC set, we can infer that its LLM-generated label happens to be the same as the noisy assigned label, which is only relatively probable when the number of classes is small and the dataset is difficult. Therefore, the assumption **can be satisfied** in most cases. To verify this, we report the EC accuracy in truly-noised datasets (SemEval, TREC, and Hausa).
>
>
>
> Acc| Easy Clean (EC) | Hard Clean (HC)| True Noisy (TN)| Overall
> |----------|-------------|------------|-----|------|
>  SemEval| 96.94% | 0.53% |57.69%| 63.35%
>  TREC| 97.20%| 1.67% |47.45% |68.76%
> Hausa| 95.65%| 2.41% | 46.12% | 50.17%
>
>
>
> The results show that the LLM accuracy is still **high in most cases**. Nevertheless, the accuracy is relatively low on Hausa, as this dataset is significantly more difficult and also with only five classes. Note that according to the definition of the HC (hard clean) set, the LLM accuracy should be relatively low on these samples. Thus, in our framework, we do not directly use LLM-generated confidences, but instead use PLM-generated confidences for fine-tuning.
>
>
>
>
> Regarding the percentages of the three categories in experiments, we present the detailed statistics of datasets with 20% Symmetric Noise as follows. For brevity, we only report datasets 20Ng and AGNews here, and will include all detailed statistics in the appendix in the revised version.
>
>
>
> | 20Ng| Easy Clean (EC) | Hard Clean (HC)| True Noisy (TN)|
> |:----------:|:-------------:|:-------------:|:------:|
> | Groundtruth| 60.91% (5,513)|19.79% (1,791) |19.30% (1,747)|
> | Percentage | 60.91% (5,513) | 18.80% (1,702) | 20.29% (1,836) |
>
>
> |AGNews|Easy Clean (EC) | Hard Clean (HC)| True Noisy (TN)|
> |:----------:|:-------------:|:-------------:|:------:|
> | Groundtruth| 65.31% (26,124)|13.60% (5,440) |21.09% (8,436) |
> | Percentage | 65.31% (26,124) | 13.04% (5,217) | 21.64% (8,659) |
>
>
> Note that the ideal separation of the EC set is always identical to the separation in practice, as this process does not require access to the true labels.
>
> From the statistics, we can observe that the TN set ratio is very close to the real noise ratio (20%), which indicates that our framework can **precisely separate noisy samples**. The ratios of EC and HC sets are also close to the real ratios, denoting that a more precise separation helps learning from noisy labels.
>
> &nbsp;
>
>
> > Q1: In Figure 4, LAFT with 0.9 ratio noise from AN or IDN obtains nearly 60% accuracy. Why the accuracy under these settings do not have a significant degradation?
>
> A1: Thank you for pointing out the performance of LAFT under high noise ratios. We achieve this by leveraging the confidences provided by LLMs. As a result, the performance also benefits from the inherent property of LLMs, which do not require training and are thus more **robust to noisy data**. On the other hand, existing baselines appear to struggle with large noisy ratios as they need to be trained on selected clean data, which can be hard to achieve when a large noisy ratio is present.
>
> &nbsp;
>
>
> > Q2: What happens when the noise ratio becomes 0, but the framework doesn't know this prior ratio while keeping the assumption that there are noises in the dataset?
>
> A2: Thank you for posing an insightful question about the framework's behavior in the absence of any noise. Since our work is proposed for noisy scenarios, we do not include results on totally clean datasets, as in existing works for learning from noisy labels [1,2,3].
>
> When the noise ratio becomes 0, instead of separating a large TN set (as the number of identified noisy samples is small), our framework still separates EC and HC sets, because these two sets exist even in totally clean data. With samples in these two sets, our designed loss in Eq. (10) can still help fine-tune the PLM with the guidance from LLM. As such, our framework **retains its efficacy** when on datasets without any noise.
>
> [1] Liu, Sheng, et al. "Early-learning regularization prevents memorization of noisy labels." In NeurIPS 2020.
> [2] Englesson, Erik, and Hossein Azizpour. "Generalized jensen-shannon divergence loss for learning with noisy labels." In NeurIPS 2021.
> [3] Zhang, Yivan, Gang Niu, and Masashi Sugiyama. "Learning noise transition matrix from only noisy labels via total variation regularization." In ICML, 2021.

---

### Meta-Review · Area_Chair_GbgR · 2023-09-11

**Recommendation:** 3

**Metareview:**

The paper addresses the problem of noisy training data for fine-tuning pre-trained language models and proposes a method that leverages external large language models. In particular, the paper separates the training examples into three parts (easy clean, hard clean, and true noisy) and utilizes different objectives to learn from those different categories.

Pros / Strengths:
- The authors conduct exhaustive experiments on both synthetic and real-world noisy datasets.
- The paper is well written and the method is well described.
- Mitigating noise in data is of general interest and any significant contribution is likely to have wide impact.

Cons / Weaknesses:
- The method assumes the existance of high-quality large language models.
- The relationship of the method (and empirical comparisons) to other areas in NLP with noisy data labeling or other LLM-based methods is missing.

Action items for improving the paper:
- The comparisons with knowledge distillation as presented during the author rebuttal phase are an interesting baseline and should be added to the paper to contrast the work against this line of work.
- The related work section should be extended, i.a., with work on label smoothing

---

### Decision · Program_Chairs · 2023-10-07

**Decision:**

Accept-Findings

**Comment:**

The paper addresses the problem of noisy training data for fine-tuning pre-trained language models and proposes a method that leverages external large language models. In particular, the paper separates the training examples into three parts (easy clean, hard clean, and true noisy) and utilizes different objectives to learn from those different categories.

Pros / Strengths:
- The authors conduct exhaustive experiments on both synthetic and real-world noisy datasets.
- The paper is well written and the method is well described.
- Mitigating noise in data is of general interest and any significant contribution is likely to have wide impact.

Cons / Weaknesses:
- The method assumes the existance of high-quality large language models.
- The relationship of the method (and empirical comparisons) to other areas in NLP with noisy data labeling or other LLM-based methods is missing.

Action items for improving the paper:
- The comparisons with knowledge distillation as presented during the author rebuttal phase are an interesting baseline and should be added to the paper to contrast the work against this line of work.
- The related work section should be extended, i.a., with work on label smoothing